# Memory Effects, Multiple Time Scales and Local Stability in Langevin Models of the S&P500 Market Correlation

**DOI:** 10.3390/e25091257

**Published:** 2023-08-24

**Authors:** Tobias Wand, Martin Heßler, Oliver Kamps

**Affiliations:** 1Institut für Theoretische Physik, Westfälische Wilhelms-Universität Münster, 48149 Münster, Germany; m_hess23@uni-muenster.de; 2Center for Nonlinear Science Münster, Westfälische Wilhelms-Universität Münster, 48149 Münster, Germany; okamp@uni-muenster.de

**Keywords:** Langevin equation, econophysics, Bayesian estimation, memory effects, non-Markovian dynamics

## Abstract

The analysis of market correlations is crucial for optimal portfolio selection of correlated assets, but their memory effects have often been neglected. In this work, we analyse the mean market correlation of the S&P500, which corresponds to the main market mode in principle component analysis. We fit a generalised Langevin equation (GLE) to the data whose memory kernel implies that there is a significant memory effect in the market correlation ranging back at least three trading weeks. The memory kernel improves the forecasting accuracy of the GLE compared to models without memory and hence, such a memory effect has to be taken into account for optimal portfolio selection to minimise risk or for predicting future correlations. Moreover, a Bayesian resilience estimation provides further evidence for non-Markovianity in the data and suggests the existence of a hidden slow time scale that operates on much slower times than the observed daily market data. Assuming that such a slow time scale exists, our work supports previous research on the existence of locally stable market states.

## 1. Introduction

The S&P500 is an aggregated index of the stocks of the 500 largest companies traded at US stock exchanges and therefore serves as an indicator of the overall US economic performance. Estimating and predicting the correlation between different assets is crucial for optimal portfolio selection and risk management and has been the focus of financial research since Markowitz’s portfolio theory [1].

The economy is a system with a large number of nonlinearly interacting parts, such as, e.g., traders, industrial producers, consumers and employees. These interactions can lead to positive and negative feedbacks and emergent effects like self-organisation. Therefore, the economy can be understood as a complex system and is amenable to the analysis tools from complex systems science [2,3].

Due to the increasing availability of data for the economy, highly data-driven methods can be applied in this field of research and many researchers have chosen to focus on the correlations between the relative price changes of different stocks, i.e., the correlations of the stocks’ returns [1,4,5,6,7,8]. In [9,10,11], the authors identified states of the economy by analysing correlation matrices of daily S&P 500 data and found eight clusters. These can be interpreted as economic states reflecting, e.g., times of economic growth or crises [9] and are found to be locally stable [10,11]. Further analyses showed that exogenous events, precursors for crises and collective market behaviour can also be identified via the correlation matrix approach [12,13,14]. Analysis of the eigenvalues of the correlation matrix revealed that the largest eigenvalue, called the “market mode” or “market factor”, is far outside of the expected spectrum of purely random matrices and is furthermore essentially captured by the mean market correlation. The market mode is thus interpreted as a hint at collective behaviour in the economic system [5,8] and used to classify market states by focusing on the market mode [11]. Moreover, a study on how the market mode’s eigenvector entries scale with the length of the time series revealed a strongly coupled market core and weakly linked periphery [3]. Thus, the mean market correlation or market mode captures a lot of information about the financial market in a low-dimensional observable and therefore, we analyse the time series of the mean market correlation to gain new insights into the market dynamics.

Ever since Bachelier’s seminal work introduced the random walk model to describe stock prices [15], the inherent stochasticity of financial data has been taken into account by researchers and practitioners alike. The Langevin equation is a model for stochastic differential equations that includes a deterministic drift and a random diffusion and can be used to describe such stochastic data. Much research has been devoted to estimating Langevin equations from data [16,17,18,19,20,21,22] and Langevin equations have found applications in various fields of research such as fluid dynamics [23], molecular dynamics [24] and meteorology [25] (cf. [26] for a review of applications). The generalised Langevin equation (GLE) expands the Langevin model by introducing a kernel function to include memory effects [27]. Bayesian statistics includes a collection of methods that reverse the classical approach to statistics and focuses on calculating posterior distributions of model parameters as probabilities conditional on the observed data [28,29]. This approach enables an efficient estimation of non-Markovian Langevin equations [30,31]. Also, other time series analysis methods can be implemented in the Bayesian framework to, e.g., detect change points in time series data [32].

We show that an estimated GLE model manages to achieve a high goodness-of-fit for the correlation time series and implies a strong memory effect, which has to be taken into account when predicting future market correlations. Furthermore, we perform a detailed comparison of a Markovian mono-time scale and a non-Markovian two-time scale model with a hidden slow time scale, which confirms the GLE analysis results regarding non-Markovian memory effects. Additionally, the analysis supports the theory of locally stable and quasi-stationary market states discussed in Stepanov et al. [11] and provides evidence that a hidden slow time scale might be involved in the mean market correlation dynamics. It is not far-fetched to assume that a complex system like the human economy involves a multitude of time scales and our findings coincide with such reasoning. The slow time scales might be connected to business and innovation cycles or similar economic mechanisms even though we could not extract a quantitatively reasonable magnitude of the hidden slow time scale.

The remainder of this article is structured as follows: Section 2 explains the data gathering and preprocessing in Section 2.1, the Bayesian methodology in Section 2.2, the GLE estimation procedure in Section 2.3 and the resilience analysis in Section 2.4. Section 3 describes the goodness-of-fit and the estimated memory parameters for the GLE model in Section 3.1 as well as the resilience results for two time scales in Section 3.2. Finally, the results of these analyses are interpreted and compared to the results from [11] in Section 4.

## 2. Data and Methods

### 2.1. Data Preparation

The S&P 500 is a stock index containing 500 large US companies traded at American stock exchanges. Daily stock data from these companies were downloaded via the Python package *yfinance* [33] for the time period between 1992 and 2012, which is the same time period as in [10,11]. Although this limits our analysis to slightly outdated data, it ensures that we can compare our methodology and results to the findings in [11]. Only stock data of companies that were part of the S&P 500 index during at least 99.5% of the time were used for this analysis. Note that while this data selection is in line with [9,10,11,34], it introduces a survivorship bias by disregarding the information about companies that went bankrupt during this time period, as pointed out in [34]. However, it ensures that all considered companies experience the same macroeconomic events, which would not be the case for those companies that vanish during the considered time period due to bankruptcy. If a company’s price time series $P_{t}$ is not available for the full time period, the remaining 0.5% of the price time series data are interpolated linearly with the *.interpolate()* method in *pandas* [35,36]. Overall, there are 249 companies’ time series for 5291 trading days with one observation per date. Each company’s returns $R_{t}=\left(P_{t+1}-P_{t}\right)/P_{t}$ are locally normalised to remedy the impact of sudden changes in the drift of the time series with the method introduced in [37], as
(1)$$r_{t}=\frac{R_{t}-{\langle R_{t}\rangle }_{n}}{\sqrt{{\langle R_{t}^{2}\rangle }_{n}-{\langle R_{t}\rangle }_{n}^{2}}}.$$
Here, ${\langle \cdots \rangle }_{n}$ denotes a local mean across the *n* most recent data points, i.e., $r_{t}$ is subjected to a standard normalisation transformation with respect to the local mean and standard deviation (i.e., volatility σ). Following [9], n=13 was chosen for the daily data. For each time step *t* and each pair of sectors i,j the local correlation coefficients
(2)$$C_{i,j}=\frac{{\langle r_{t}^{\left(i\right)}r_{t}^{\left(j\right)}\rangle }_{\tau }-{\langle r_{t}^{\left(i\right)}\rangle }_{\tau }{\langle r_{t}^{\left(j\right)}\rangle }_{\tau }}{\sigma_{\tau }^{\left(i\right)}\sigma_{\tau }^{\left(j\right)}}$$
are calculated over a time period of τ=42 trading days such as in [9] (the 42 working days correspond to 2 trading months) with the local standard deviations $\sigma_{\tau }^{\left(i\right)}$. It should be kept in mind that the correlation coefficient only detects linear dependencies. However, it is used ubiquitously in data-driven inference on financial markets [3]. As shown via Principle Component Analysis in [11], the mean correlation
(3)$$\bar{C}=\frac{1}{N}\sum_{i,j}C_{i,j}$$
already describes much of the variability in the data as the largest eigenvalue or market mode of the correlation matrix. While the majority of the eigenvalue spectrum follows the Marcenko-Pastur distribution that is expected for purely random matrices, the market mode is far outside of this spectrum. Therefore, it hints at a strongly non-random effect [2,3,5]. Note that [3] also points out that this property is not only found in US stock markets, but is a general observation across markets. Hence, it makes sense to restrict the analysis to this one-dimensional time series $\bar{C}\left(t\right)$ (shown in Figure 1). The time *t* is here selected as the central value of the time window of length τ in order to have a symmetrical window. The preprocessed time series is available via [38].

### 2.2. Bayesian Statistics

Bayes’ theorem for the conditional probability distributions f(·|·) of model parameters θ and observed data
(4)$$f_{post}\left(\theta |x\right)∼f_{prior}\left(\theta \right)f\left(x|\theta \right)$$
connects the standard statistical likelihood f(x|θ) and prior knowledge about the model parameters $f_{prior}$ to a posterior distribution $f_{post}$ of the unknown model parameters conditional on the observed data [39]. A bundle of methods have been derived from this approach and are collectively referred to as Bayesian Statistics [28,29]. Markov chain Monte Carlo algorithms (MCMC) can be used to infer the posterior distribution by simulating several random walkers that explore the (potentially high-dimensional) posterior distribution [40]. It generates uncorrelated samples of the model parameters by cutting off the first $n_{burn}$ exploration steps as a burn-in period and thinning the remaining samples. The resulting MCMC samples can also be used to integrate the posterior distribution over all but one model parameter $\theta_{i}$ to derive the marginal posterior distribution $f(\theta_{i}|x)$ of this single parameter. Parameter estimation can then be conducted via the mean of the posterior distribution or via its maximum (abbreviated as MAP for maximum a posteriori) and credible intervals (CIs) of, e.g., 95% credibility can be directly derived from the quantiles of the samples for $\theta_{i}$.

### 2.3. Fitting a Generalised Langevin Equation with Memory Kernel

Sections 4 and 5 of [11] analyse the one-dimensional mean correlation time series by fitting a Langevin model
(5)$$\frac{d\bar{C}}{dt}\left(t\right)=D^{\left(1\right)}\left(\bar{C},t\right)+\sqrt{D^{\left(2\right)}\left(\bar{C}\right)}\Gamma \left(t\right)$$
with independent Gaussian noise Γ, a deterministic time-dependent drift function $D^{\left(1\right)}$ and a time-independent diffusion function $D^{\left(2\right)}$. Note that the Langevin equation is a Markovian model, i.e., it has no memory. As an alternative model, a Generalised Langevin Equation (GLE) includes previous values of the time series in a memory kernel K, but has only time-independent parameters with a functional equation
(6)$$\frac{d\bar{C}}{dt}\left(t\right)=D^{\left(1\right)}\left(\bar{C}\right)+\int_{s=0}^{t}K\left(s\right)\bar{C}\left(t-s\right)ds+\sqrt{D^{\left(2\right)}\left(\bar{C}\right)}\Gamma \left(t\right).$$

A Bayesian estimation of the parameters of Equation (Equation 6) is implemented in [30]. It discretises the memory kernel K(k) to $K_{k}$ and discretises the observed data into $n_{B}$ different bins to significantly speed up the estimation procedure. Because an overlap of the windows used to calculate $\bar{C}$ can lead to artefacts in the memory effects (cf. Appendix A), we choose to calculate $\bar{C}$ with different parameters than τ=42 and s=1 from [11] (the mean correlation of every *s*th day is retained for the analysis; hence, for s=1, the full time series is used). Instead, we set τ=5 and s=5, meaning that the mean market correlation $\bar{C}$ is calculated over one trading (τ=5) week for the end of the trading week in disjoint windows (i.e., we create a time series whose length is 1/s=20% of the original time series). Because using disjoint intervals automatically leads to a thinning of the time series, this seemed like a useful trade-off with a real-world interpretation as weekly correlation. It is worth noting that while the calculation of $\bar{C}$ only considers data from five trading days, the correlation matrix $C_{i,j}$ encompasses approximately $10^{4}$ correlation values. Even though each (i,j) correlation pair is determined over a brief time window, the large ensemble of $C_{i,j}$ values, with their average being $\bar{C}$, underscores our confidence in the accuracy of the mean $\bar{C}$. The resulting time series is shown in Figure 1 together with the time series used in [11] and although it obviously becomes much noisier due to the shorter window size, it still retains some of the features of the less noisy time series such as, e.g., the position of spikes with high correlation.

As described in [30], the memory effects are aggregated to a quantity $K_{k}$ (named $\kappa_{k}$ in [30]) that measures the strength of all memory effects from 1 up to *k* time steps ago. If $K_{k}$ saturates towards a plateau at step $k_{0}$, then $k_{0}$ is the maximum length of any reasonable memory kernel. This often also coincides with kernel values of $K_{k_{0}}\approx 0$ at $k_{0}$. The estimated values for the model parameters are chosen by calculating the marginal posterior distribution and choosing the mean or MAP parameter estimation. The goodness-of-fit is also evaluated by using MCMC to simulate an artificial time series ${\bar{C}}_{t}^{\left(a\right)}$ with the estimated model and comparing the autocorrelation function and the distributions of ${\bar{C}}_{t}^{\left(a\right)}$ and the increments ${\bar{C}}_{t}^{\left(a\right)}-{\bar{C}}_{t-j}^{\left(a\right)}$ to those of the original data. Finally, the inferred model structure can be trained on only a subset of the data (training data) and be used to predict the remaining test data to evaluate its predictive performance.

### 2.4. Resilience Estimation

By applying Bayes’ theorem (Equation 4), we can deduce a quantitative measure of resilience under given model assumptions. Therefore, we infer the parameters of two models of stochastic differential equations in a rolling window approach that allows us to resolve the time evolution of the resilience and noise level and accounts for the quasi-stationary nature of the time series that is observed in Stepanov et al. [11]. Following the quasi-stationarity argument, we assume to be in a fixed state in each window, i.e., the fixed point ${\bar{C}}^{*}$, which is approximated by averaging over the mean market correlation data per window. First, we estimate a Markovian Langevin Equation (Equation 5) with the Taylor-expanded drift
(7)$$\begin{array}{cc} D_{\bar{C}}^{\left(1\right)}\left(\bar{C}\left(t\right),t\right) & =\alpha_{0}\left(t\right)+\alpha_{1}\left(t\right)\left(\bar{C}-{\bar{C}}^{*}\right)+\alpha_{2}\left(t\right){\left(\bar{C}-{\bar{C}}^{*}\right)}^{2}+\alpha_{3}\left(t\right){\left(\bar{C}-{\bar{C}}^{*}\right)}^{3}+O\left({\bar{C}}^{4}\right)\\ \; & =\theta_{0}\left(t;{\bar{C}}^{*}\right)+\theta_{1}\left(t;{\bar{C}}^{*}\right)\;\cdot \;\bar{C}+\theta_{2}\left(t;{\bar{C}}^{*}\right)\;\cdot \;{\bar{C}}^{2}+\theta_{3}\left(t;{\bar{C}}^{*}\right)\;\cdot \;{\bar{C}}^{3}+O\left({\bar{C}}^{4}\right) \end{array}$$
with the drift parameters $\theta_{0,\ldots ,3}\left(t;{\bar{C}}^{*}\right)\equiv \theta_{0,\ldots ,3}\left({\bar{C}}^{*}\right)$ per rolling window and the constant diffusion $D^{\left(2\right)}\left(\bar{C}\right)=\theta_{4}^{2}\equiv const.=:\sigma^{2}$. We choose uninformed priors, which are given by
(8)$$\begin{array}{c} p_{\mathrm{prior}}\left(\theta_{0},\theta_{1}\right)=\frac{1}{2\pi {\left(1+\theta_{1}^{2}\right)}^{\frac{3}{2}}} \end{array}$$
for the linear part of the drift function and the Jeffreys scale prior [29]
(9)$$\begin{array}{c} p_{\mathrm{prior}}\left(\theta_{4}\right)=\frac{1}{\theta_{4}} \end{array}$$
for the noise level $\theta_{4}$ with suitable prior ranges. The priors of higher-order parameters are chosen to be
(10)$$\begin{array}{cc} \; & p_{\mathrm{prior}}\left(\theta_{2}\right)=N\left(\mu =0,\tilde{\sigma }=4\right)\;\mathrm{and}\; \end{array}$$
(11)$$\begin{array}{cc} & p_{\mathrm{prior}}\left(\theta_{3}\right)=N\left(\mu =0,\tilde{\sigma }=8\right) \end{array}$$
with Gaussian distributions N centred around the mean μ=0 with a standard deviation $\tilde{\sigma }=4$ and $\tilde{\sigma }=8$.

Since we consider economic systems to operate on multiple time scales whose combination often leads to non-Markovian time series, we additionally introduce a two-dimensional non-Markovian model analogous to Willers and Kamps [22]. This model is given by
(12)$$\begin{array}{cc} \frac{d\bar{C}\left(t\right)}{dt} & =D_{\bar{C}}^{\left(1\right)}\left(\bar{C},t\right)+\sqrt{D_{\bar{C}}^{\left(2\right)}\left(\bar{C},t\right)}\;\cdot \;\lambda \end{array}$$
(13)$$\begin{array}{cc} \frac{d\lambda \left(t\right)}{dt} & =D_{\lambda }^{\left(1\right)}\left(\lambda ,t\right)+\sqrt{D_{\lambda }^{\left(2\right)}\left(\lambda ,t\right)}\;\cdot \;\Gamma \left(t\right) \end{array}$$
with a hidden Ornstein–Uhlenbeck process (OU-process) λ, drift $D_{\lambda }^{\left(1\right)}\left(\lambda ,t\right)=-\frac{1}{\theta_{5}^{2}}\;\cdot \;\lambda$ and diffusion $D_{\lambda }^{\left(2\right)}\left(\lambda ,t\right)=\frac{1}{\theta_{5}^{2}}$. Drift $D_{\bar{C}}^{\left(1\right)}\left(\bar{C},t\right)$ and diffusion $D_{\bar{C}}^{\left(2\right)}\left(\bar{C},t\right)$ of the observed process $\bar{C}\left(t\right)$ remain unchanged. The non-Markovian analogue to the constant noise level $\sigma^{2}$ of the Langevin equation is given by the composite noise level
(14)$$\begin{array}{c} \Psi =\sqrt{D_{\bar{C}}^{\left(2\right)}\left(\bar{C},t\right)}\;\cdot \;\sqrt{D_{\lambda }^{\left(2\right)}\left(\lambda ,t\right)}\;\cdot \;h \end{array}$$
with the small discrete sampling time step *h*.

For the OU-process, an invariant prior of a straight line and a scale prior for the diffusion are multiplied:(15)$$\begin{array}{c} p_{\mathrm{prior}}\left(\theta_{5}\right)=\frac{\theta_{5}}{2\pi {\left(1+{\left(-\frac{1}{\theta_{5}}\right)}^{2}\right)}^{\frac{3}{2}}}. \end{array}$$
Furthermore, via the prior we introduce a pre-defined time scale separation of the time scales $\tau_{\bar{C}}$ and $\tau_{\lambda }$ of the observed and unobserved process, respectively; i.e., we require either $\tau_{\bar{C}}>\gamma \;\cdot \;\tau_{\lambda }$ or $\tau_{\lambda }>\gamma \;\cdot \;\tau_{\bar{C}}$ with a scale separation coefficient γ. The characteristic time scales [41] are approximated by
(16)$$\begin{array}{c} \tau_{\nu }={\left|\left.{\left(\frac{dD_{\nu }^{\left(1\right)}\left(\nu ,t\right)}{d\nu }\right)}^{-1}\right|\right|}_{\nu =\nu^{*}}\;\;\mathrm{with}\;\nu \in \left\{\bar{C},\lambda \right\}. \end{array}$$
The priors for the model of the observed data $\bar{C}\left(t\right)$ also remain unchanged with the exception of the term $D_{\bar{C}}^{\left(2\right)}\left(\bar{C},t\right)=\theta_{4}^{2}$, which corresponds to a coupling constant in the non-Markovian model. We simply apply a Gaussian prior similar to the one in Equation (Equation 10) to $\theta_{4}$ in this case.

Inspired by the formalism of linear stability analysis for both models, we calculate the drift slope
(17)$$\zeta ={\left.\frac{dD_{\bar{C}}^{\left(1\right)}\left(\bar{C}\right)}{d\bar{C}}\right|}_{\bar{C}={\bar{C}}^{*}}$$
per window as a Bayesian parametric resilience measure. For the Markovian model, the calculations are performed with the open-source Python toolkit *antiCPy* [42,43]. In this modelling framework, a stable state corresponds to a negative drift slope ζ, whereas a changing sign indicates destabilisation via a bifurcation. More details on the procedure can be found in [32].

## 3. Results

### 3.1. Estimated GLE Model

The data are split into $n_{B}=10$ equally wide bins and an initial modelling attempt with a rather long kernel $k_{\max}=10$ is tested, i.e., $K_{q}=0$ for all $q>k_{\max}$. It shows a plateau emerging at around k≥6 (cf. Appendix B). Hence, a model with $k_{\max}=6$ is used as a reasonable length for the memory kernel and its goodness-of-fit is evaluated. We then use a very conservative estimation of the memory up to $k_{\max}=3$ to evaluate the GLE’s predictive power.

#### 3.1.1. Goodness-of-Fit

A time series of length $10^{5}$ is simulated via Euler–Maruyama integration of the GLE model to compute the autocorrelation function (ACF) and to compare it to the ACF of the original time series. The best model is chosen via MAP estimation in the Bayesian framework of [30], but selecting the mean estimation yields almost identical results. Both ACFs are computed via the function *statsmodels.tsa.stattools.acf* from the Python package *statsmodels* [44]. Figure 2 shows that the two ACFs show very good agreement up to lags of 10 trading weeks and decent agreement up to lags of 20 weeks. The distributions of the time series and the first two increments for the original and the simulated data are shown in Figure 3 for the MAP estimation and also for the mean estimation. Figure 3 shows an almost perfect overlap between the two estimation procedures and a good agreement between the estimated time series and the original time series, especially for the increment distributions. Overall, these diagnostics indicate that the estimated model with memory kernel length 6 manages to reproduce these important statistical properties of the original time series.

Because the realised values of the time series $\bar{C}$ are not distributed uniformly, we also use a modelling procedure with unequal bin widths so that each of the 10 bins has the same amount of data. However, this barely changes the model diagnostics shown in Figure 2 and Figure 3. The only noticeable change is that for long kernels with length ≥10, the ACF seems to be captured a little more poorly in the shown range of Figure 2, but manages to fit more closely to the empirical ACF for large lags at around r≈100. While the regular LE without memory has a very similar increment distribution, the ACF is considerably poorer than the GLE.

#### 3.1.2. Estimated Memory Kernel

To make further inference on the memory kernel and to estimate its quantitative effect, the posterior distribution of the model with memory length 6 is sampled via MCMC. 100 walkers are simulated for $10^{5}$ time steps and after a burn-in period of 450 initial time steps is discarded, the chains are thinned by only keeping every 450th step to obtain uncorrelated samples. Bayesian CIs can now be calculated at the 95% level for each parameter in the kernel function and the results are shown in Figure 4.

The CIs of $K_{4}$ are already very close to the zero line and those of $K_{5}$ include zero, meaning that there is no evidence for a memory term at five weeks distance. Interestingly, the CIs for $K_{6}$ clearly exclude the value zero. Because it is difficult to exactly identify the beginning of the plateau in the memory aggregation in Figure A2, it may be possible that the plateau already emerges at k=5 and that the nonzero memory kernel $K_{6}$ is therefore unreliable. However, the results in Figure 4 clearly imply a nonzero memory effect for four time steps with a clearly nonzero effect strength for memories up to k=3 trading weeks.

#### 3.1.3. Prediction via the GLE with Kernel Length 3

With a conservative interpretation of the results in Section 3.1.2, a model with memory length k=3 is used to evaluate the GLE’s power to predict a future value $y_{t+1}$ by forecasting an accurate prediction ${\hat{y}}_{t+1}$. It is tested against a regular Langevin equation (LE) model without any memory effects (corresponding to k=0 and also estimated with the code in [30]) and against the naive benchmark of predicting the next time step $y_{t+1}$ by simply setting it to the last previously known value: ${\hat{y}}_{t+1}=y_{t}$. To test these three methods, the Langevin models are trained on the first α% of the time series (the training data) and the predictions of the GLE, the LE and the naive forecast are evaluated on both the training data (as in-sample predictions) and on the remaining 1−α% test data (as out-of-sample predictions). The coefficient of prediction $\rho^{2}$ is used to evaluate their predictive accuracy. If *n* observations $y_{1,\ldots ,n}$ are forecasted as ${\hat{y}}_{1,\ldots ,n}$, then the coefficient of prediction is given by
(18)$$\rho^{2}=1-\frac{\sum_{i=1}^{n}{\left({\hat{y}}_{i}-y_{i}\right)}^{2}}{\sum_{i=1}^{n}{\left(\bar{y}-y_{i}\right)}^{2}}$$
with the $\bar{y}$ denoting the mean value. It takes the value of $\rho^{2}=1$, if the prediction is always exactly true, and $\rho^{2}=0$, if the prediction is only as accurate as always using the mean $\bar{y}$, and $\rho^{2}<0$, if it is less accurate than using the mean and is computed via the function *sklearn.metrics.r2_score* from [45]. The results in Table 1 show that the GLE model consistently achieves the highest accuracy on in-sample and out-of-sample predictions for the three chosen test data sizes. Notably, the negative $\rho^{2}$ of the naive method for the test data indicates that the out-of-sample prediction is by no means trivial, meaning that the low, but positive $\rho^{2}$ of the GLE on the test data nevertheless constitutes a good performance. The GLE achieves slightly better results than the LE, indicating that the memory effect should be taken into account for prediction tasks. Figure 5 shows the predictions of the LE and GLE for the in-sample and out-of-sample predictions with α=90%. The same visual comparison between naive forecast and the GLE can be found in the Appendix D.

### 3.2. Hidden Slow Time Scale and Non-Markovianity

Applying the resilience analysis method from Section 2.4, we can deduce some interesting evidence for multiple time scales and further confirm that non-Markovianity is present in the considered economic time series. The results for both the simple Markovian and multi-scale non-Markovian model (cf. (Equation 5) and (Equation 12), respectively) are compared in Figure 6a–c. The parameters of the calculations are listed in Table A1 of Appendix E.

Stepanov et al. [11] argue for quasi-stationary economic states, which are occupied over finite time periods before they transition into another quasi-stationary economic state. The estimated potential landscape for the mean market correlation in the article [11] suggests that there might be shifts in the fixed point positions over time, but no bifurcation-induced tipping (B-tipping) is involved, i.e., no qualitative change of stable and unstable fixed points or attractors is observed. Instead of B-tipping mechanisms, the intrinsic economic stochasticity drives the jumps between alternative quasi-stationary states, which is a mechanism basically related to noise-induced tipping (N-tipping).

If we choose a model that captures the key features of the data generating process, we should thus be able to uncover generally negative drift slope estimates $\hat{\zeta }$, corresponding to data of a locally quasi-stationary state in each rolling window of the mean market correlation $\bar{C}\left(t\right)$, the data of which is shown again in Figure 6a.

In this spirit, we find the non-Markovian model with an unobserved slow time scale, i.e., $\tau_{\lambda }>\gamma \;\cdot \;\tau_{\bar{C}}$ with γ=2, to yield the expected result of negative drift slope estimates $\hat{\zeta }$ as presented in Figure 6b and indicated by the red solid line with green credibility bands (CBs).

This result is supported by rather intuitive qualitative considerations: Economic processes are known to operate on various fast and slow time scales. On the one hand, these dynamics contain, e.g., high-frequency trading, trading psychology and sudden external events, whether that might be political, sociological, severe weather or other impacts. These fast-scale processes could be called “economic weather” in a climatologist’s metaphor. On the other hand, long-term “economic climate evolution” takes place on much slower time-scales. Examples could be innovation processes and technical revolutions such as the invention of the steam engine or the internet, economic cycle theories such as that of Marx (τ∼ 2 to 10 years) [46,47], Keynes, Schumpeter or Kondratjew [48,49,50,51], cycles of fiscal and demographic developments [52], cultural evolution [53] and generational changes influencing economic reasoning, adaptions to climate change and the scarcity of resources and much more. Keeping that in mind, the trading day resolution of the data is rather fine-grained and it might be reasonable to assume that a hidden slow time scale is present in the data. The slow time scale of the presented non-Markovian model is determined by the upper boundary of the prior range, which is chosen to correspond roughly to 6 years, which is the time of an average business cycle and coincides with the first zero-crossing of the autocorrelation function of the mean market correlation $\bar{C}\left(t\right)$. If the prior is chosen more broadly, the model converges to roughly 700 to 4000 years, which is not a plausible magnitude of the time scale of economic evolution (cf. Appendix E). However, the main results of local quasi-stationary economic states is not affected by the prior choice. The estimation of a more reasonable magnitude of the slow time scale without prior restriction might be prohibited by the very limited amount of data per window, since the window range does not even include one complete business cycle, which would be the smallest proposed slow time scale candidate. Further note that the two-time scale non-Markovian model in Equation (Equation 12) incorporates noise in the slow non-observed variable λ. In that way, it could formally reflect intermediate dynamics on a time scale $\tau_{N}$ with $\tau_{\bar{C}}<\tau_{N}<\tau_{\lambda }$. However, since it is coupled to the mean market correlation $\bar{C}\left(t\right)$ on trading day resolution, we consider the noise operating on a time scale $\tau_{N}<\tau_{\bar{C}}$.

In contrast to the multi-scale non-Markovian model, the mono-scale Markovian model cannot reflect the local quasi-stationarity of the economic states, postulated in Stepanov et al. [11], but the drift slope estimates $\hat{\zeta }$, indicated by the blue solid line with orange CBs in Figure 6b, suggest persistently latent instability with $\hat{\zeta }\approx 0$. The noise analogues σ and Ψ of the Markovian and non-Markovian model are almost identical, as depicted in Figure 6c, following the same color-coding. The noise level seems to increase over the years with a clear increase in the periods of the global financial crisis and the Euro crisis, which accounts for a higher N-tipping probability in this highly turbulent economic period. The noise plateau of roughly one window length around the end of the Asian and the beginning of the Russian financial crises around 1998 is probably the result of the outliers that are incorporated into the windows.

Since the discussed observations alone only allow for relatively weak qualitative deduction of the discussed features of multi-scaling and non-Markovianity, we additionally perform an analogous analysis on a synthetic time series *x* that shares the key features of a hidden slow time scale and non-Markovianity with the original one. In Figure 7, we provide a comparison of the per definition stationary first differences, (a) of the original mean market correlation $\bar{C}\left(t\right)$ and (b), of the synthetic time series *x*. The noise level in *x* is assumed to increase over time to mirror the noise level evolution of the mean market correlation $\bar{C}\left(t\right)$, suggested by the estimates in Figure 6c, and is adjusted to cover almost the range of the first differences in $\bar{C}\left(t\right)$. Only the positive trend of the mean market correlation visible in Figure 6a is not included in the simulations of *x*. In that way, the PDFs of the two time series’ first differences are shaped similarly, apart from the fact that the highly centered probability mass of the mean market correlation $\bar{C}\left(t\right)$ with steep tails due to rare outliers is a bit more smeared out into flatter tails of the synthetic time series *x*. More simulation details can be found in the Appendix E.

The resilience analyses of the synthetic time series *x* are shown in Figure 6d–f and are in very good agreement with the results of the original analysis in Figure 6a–c. That serves as an independent qualitative confirmation of our findings’ interpretation. Moreover, this interpretation is supported by two additional facts: First, the estimation of a Markovian model on the Gaussian kernel detrended version of the mean market correlation $\bar{C}\left(t\right)$ results in similar results to the multi-scale non-Markovian model. This strengthens our previous findings, because the detrending subtracts a non-stationary slow process suspected to be present in the data. We notice that weaker detrending leads to positive trends in the drift slope estimates $\hat{\zeta }$, which could be due to increasing distortions due to incomplete detrending of the non-stationarity. Second, we fit a multi-scale non-Markovian model with inverse time scale separation $\tau_{\bar{C}}>\gamma \;\cdot \;\tau_{\lambda }$ with γ=2 (i.e., the observed trading day time scale of the mean market correlation $\bar{C}\left(t\right)$ is considered to be at least two times slower than the hidden time scale). This leads to results similar to the *Markovian* model with and without detrending. This is an expected result, since the prior restriction of the time scale separation basically restricts the model to the Markovian case in which only the trading day time scale can be resolved apart from an even faster stochastic contribution. In other words, the prior assumption $\tau_{\bar{C}}>\gamma \;\cdot \;\tau_{\lambda }$ with γ=2 prohibits the incorporation of an unobserved slower time scale, even if it is present in the data. The discussed results are presented in more detail in Appendix E.

## 4. Discussion and Conclusions

The estimated GLE model manages to reproduce the statistical properties of the original data for the end-of-week correlations, as shown in Figure 2 and Figure 3. The estimated memory kernel parameters show that even with a highly conservative interpretation of the 95% credible level, there are clearly nonzero memory effects for memory terms for all lags as far back as a lag of three weeks. Therefore, it is advised to use a model with memory to describe the correlation of the S&P500 market, which is an improvement to the Markovian Langevin model estimated in [11]. Moreover, the GLE estimation presented in this article achieves a high goodness-of-fit for the entire time series, whereas Stepanov et al. used a time-dependent Langevin model by splitting the time series into different intervals and estimating Markovian Langevin equations for each of them. Our work shows that this procedure can be circumvented by using a model with memory of at least three trading weeks. The major advantage of our method is its possible application in predicting future market correlation: The time-dependent drift estimation in [11] has no clear or smooth functional dependence on time and therefore, little information can be inferred about future values of the correlation time series. Our method needs no time dependency, generalises over the entire time series and can be used to predict future correlation values, which can be used for portfolio risk assessment. As shown in Section 3.1.3, the memory kernel helps the GLE to achieve better prediction accuracy than the regular Langevin equation and much better results than the naive forecasting method of using the last observation as the predicted value.

Notably, the existence of memory effects in the market’s correlation structure can be interpreted in the context of volatility clustering. It is a well-known stylised fact from empirical research on financial markets that the volatility of a stock’s returns tends to cluster: periods of high volatility are often followed by periods of high volatility and vice versa for low volatility [3,54,55]. The correlation $\rho_{X,Y}$ between two asset returns $r_{X}$ and $r_{Y}$ with expectation values $\mu_{X},\mu_{Y}$ and volatilities $\sigma_{X},\sigma_{Y}$ is defined as
(19)$$\rho_{X,Y}=\frac{E\left[\left(X-\mu_{X}\right)\left(Y-\mu_{Y}\right)\right]}{\sigma_{X}\sigma_{Y}}$$
and therefore directly includes the volatility values $\sigma_{X}$ and $\sigma_{Y}$. Because the time series of volatility estimators $\sigma_{X}\left(t\right)$ shows a well-known memory effect, it is not far-fetched to assume a similar memory effect in the correlations $\rho_{X,Y}$ between two assets or, as we have discussed in this article, in the mean correlation of the market as a whole. Note that other memory effects have already been described and discussed in the literature, e.g., originating from large-scale traders splitting their strategy into many small orders over a longer period of time [56], via a cascading process along multiple time scales [3] or in the transition between market states [57].

These considerations are complemented by a resilience analysis that involves the estimation of a mono-time scale Markovian model and a two-time scale non-Markovian model. In contrast to the memoryless Markovian model, only the non-Markovian model exhibits the negative drift slopes, which are in line with the hypothesis of locally quasi-stationary economic states postulated and observed in Stepanov et al. [11]. An independent change point analysis approach also supports this view [58]. Overall, these findings provide new evidence for the existence of such locally quasi-stationary economic states and for the presence of a significant non-Markovian memory effect. The investigated stocks and time period were chosen to enable a comparison between our results and the findings in [11]. However, our methods can be used on data from different time periods and analysing data from other stock markets could lead to fruitful future work to cross-check the robustness and universality of our results. A possible candidate is the German DAX, for which the market mode has also been analysed [3].

Extending the analysis to more recent data opens up several new research questions: Is there a difference in memory effects before and after the 2020 Covid-crisis? Does globalisation lead to memory effects across the various national markets? Does the increasing ubiquity of algorithmic trading decrease the memory length by moving the market closer to its equilibrium? However, future research on large-scale data is necessary to adequately address those questions.

Interestingly, the resilience analysis yields some evidence that a second time scale, which is slower than the trading day time scale of the mean market correlation data, is involved in the underlying economic dynamics. Economic processes operate on various fast and slow time scales, e.g., intra-day trading, trading psychology and sudden external events—whether that might be political, sociologic, severe weather or other impacts—may be incorporated in the fast trading day resolution of the mean market correlation data. We refer to these fast-scale processes in terms of “economic weather” to employ a metaphor from climatology (cf. also the distinction in climate-like and weather-like tasks in [59] and the discussion of multiple time scales in [60,61]).

In contrast to the fast time scales, the long-term evolution of the “economic climate” might involve innovation processes and technical revolutions such as the invention of the steam engine or the internet, economic cycle theories like that of Marx (τ∼ 2 to 10 years), Keynes, Schumpeter or Kondratjew [46,47,48,49,50,51,52], cultural evolution [53] and generational changes influencing economic reasoning, adaptions to climate change and the scarcity of resources and much more. However, we were not able to derive an economically reasonable magnitude of the slow time scale, which we would expect to lie in the range of decades up to hundreds of years, corresponding to well-known economic cycle theories or cultural evolution processes. Instead, our applied MCMC model estimation without prior range restriction converges to a hidden slow time scale of roughly 700 to 4000 years. Nevertheless, our results suggest that there should be at least two time scales involved in the data-generating process, which is an interesting starting point for future research. Notably, it is not particularly surprising that we could not quantify the hidden time scale, since we employ a very simple model parametrisation, have only access to one variable of the high-dimensional economic state space and perform our estimation on small windows that do not even include the smallest economic cycle time scale of roughly two to ten years that typically correspond to business cycles. Note that the largest eigenvalues of the correlation matrix have already been interpreted in the context of different time scales with respect to Haken’s slaving principle: In [3], their time scale is found to be slower than that of the sub-dominant eigenvalue processes. Interestingly, our study hints at the existence of a hidden process operating on an even slower time scale. Against this background, it might be an interesting challenge for future research to develop more realistic models and estimation procedures that perform reliably under the circumstances of limited data per window and incomplete variable sets to uncover the manifold of hidden time scales in the complex system of the human economy.

## Figures and Tables

**Figure 1 entropy-25-01257-f001:**
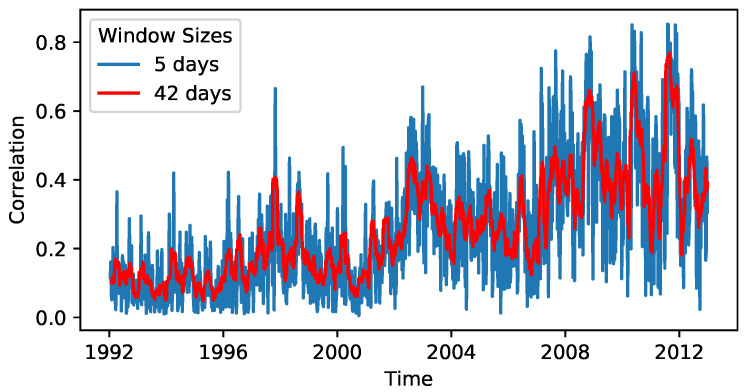
The mean correlation of the S&P500. The depicted time series are the mean values of correlation matrices, which were calculated based on moving windows of length τ days plotted against the centre of the τ-days-interval. This figure depicts τ=5 and τ=42.

**Figure 2 entropy-25-01257-f002:**
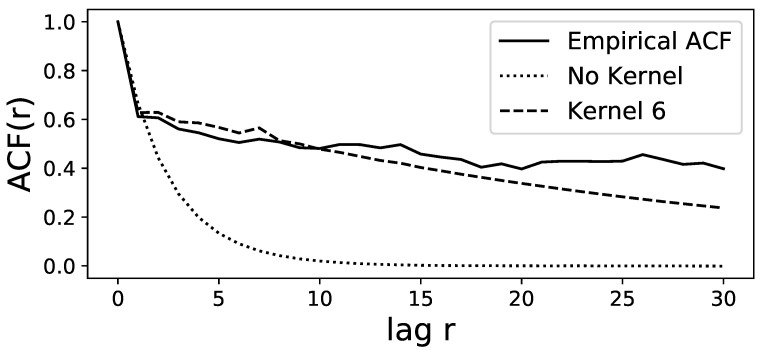
Comparison between the autocorrelation functions of the original time series (**solid line**) and the simulated data from the model with kernel length 6 (**dashed line**) simulated via MAP estimation of the MCMC-integrated marginal densities. Up to lag 30, the ACFs show good agreement. The alternative simulation via mean estimation of the density yields an almost identical ACF. However, the ACF based on the model with no memory kernel (**dotted line**) fails to capture the empirical ACF.

**Figure 3 entropy-25-01257-f003:**
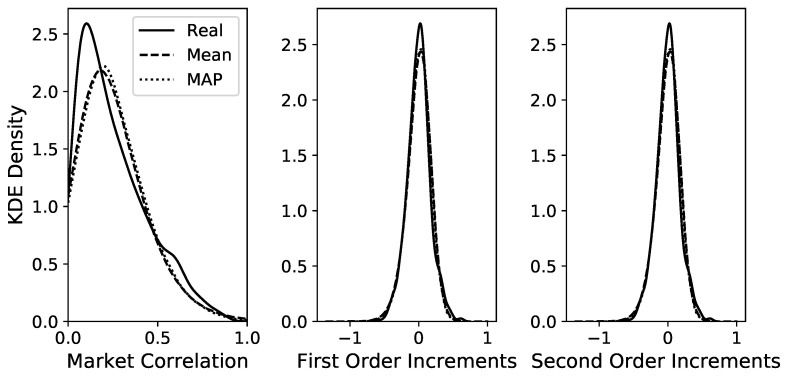
KDE plots to compare the distributions between the real data and the MCMC samples for the two estimated GLE models with kernel length 6 for the time series data ${\bar{C}}_{t}$ (**left**), the increments ${\bar{C}}_{t}-{\bar{C}}_{t-1}$ (**centre**) and ${\bar{C}}_{t}-{\bar{C}}_{t-2}$ (**right**). Differences between the two simulated models are hardly visible and overall, there is a good overlap with the real data. For the LE without memory, the respective distributions are depicted in Appendix C.

**Figure 4 entropy-25-01257-f004:**
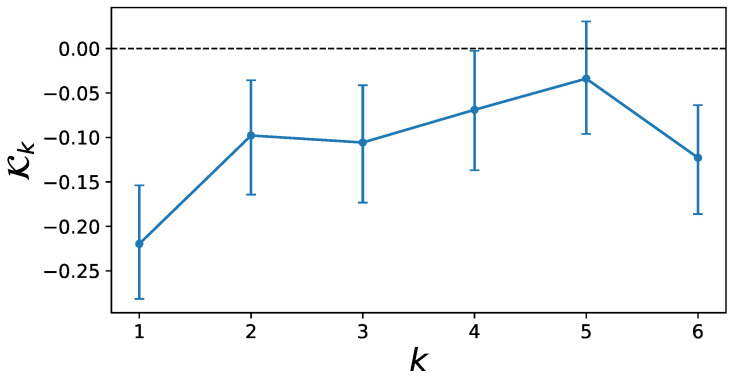
Values for the memory kernel $K_{k}$ in the model with memory length 6. Credible intervals were estimated via MCMC at the 95% level. The inclusion of 0 in the credible interval of $K_{5}$ combined with the uncertainty about the beginning of the plateau in Figure A2 imply that the nonzero memory effect for $K_{6}$ may be misleading. All previous memory kernels $K_{1,2,3,4}$ have nonzero values at the 95% credibility level.

**Figure 5 entropy-25-01257-f005:**
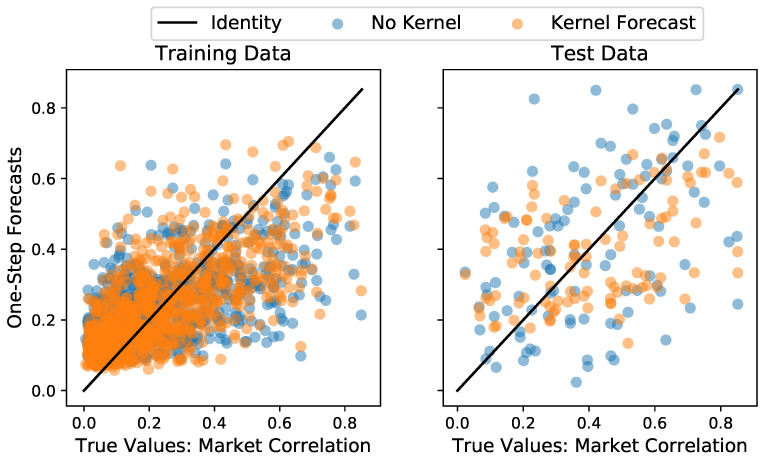
Comparison between one-step-ahead forecasts of the GLE model with Kernel against a regular Langevin estimation without memory effects on the training data (**left**) and the test data (**right**). The identity f(x)=x is given as a benchmark for perfect predictive accuracy. Here, α=90% of the time series were used as training data.

**Figure 6 entropy-25-01257-f006:**
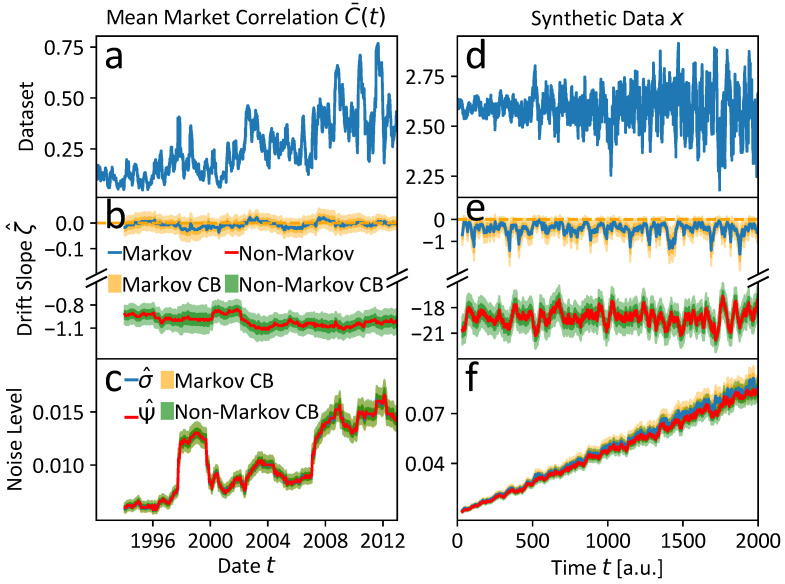
Results of the resilience analyses based on the Markovian Langevin Equation (Equation 5) and the non-Markovian model (Equation 12) with an unobserved slow time scale $\tau_{\lambda }$, i.e., under the prior assumption $\tau_{\lambda }>2\;\cdot \;\tau_{\bar{C}}$: (**a**) for the mean market correlation $\bar{C}\left(t\right)$ and (**d**) for a synthetic dataset *x* that shares the multi-scaling features and non-Markovianity with the economic time series $\bar{C}\left(t\right)$. (**b**) The Markovian model suggests persistent latent instability, i.e., $\hat{\zeta }\approx 0$, whereas the non-Markovian slopes agree with locally quasi-stationary economic states as observed in Stepanov et al. [11]. (**c**) The noise level estimates are almost identical and tend to increase over time, which might reflect more turbulent economic times towards the the financial crisis 2008. The plateau of ca. one rolling window length around 1998 might be due to the incorporation of some outliers in the time of the ending Asian crisis and the beginning of the Russian financial crisis. (**e**) Drift slope results of the synthetic dataset *x* analogous to (**b**). The qualitative results are in good agreement with (**b**), which might be a hint that the mean market correlation $\bar{C}\left(t\right)$ exhibits non-Markovian features and is additionally governed by processes on a slower time scale $\tau_{\lambda }$. (**f**) The estimated noise levels of the synthetic dataset *x* are almost identical, which is also observed for the noise levels of the mean market correlation $\bar{C}\left(t\right)$ in (**c**). For more details see the running text.

**Figure 7 entropy-25-01257-f007:**
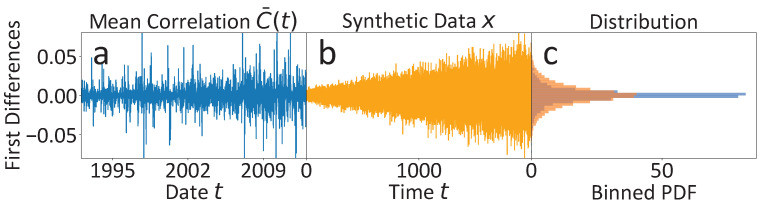
First differences (FD) and corresponding PDFs of the raw mean market correlation data $\bar{C}\left(t\right)$ and the synthetic dataset *x*, which shares the key features of non-Markovianity and an unobserved slow time scale with the mean market correlation data $\bar{C}\left(t\right)$. (**a**) The FDs of the mean market correlation are highly centered around zero with some outliers. (**b**) The FD capture essentially the same range as the FDs of $\bar{C}\left(t\right)$, but are less centered around zero. The time series is modelled with increasing coupling strength to imitate the increase of the noise level found for the mean market correlation $\bar{C}\left(t\right)$ in Figure 6c. (**c**) The FD PDFs of both time series are comparable. The PDFs only deviate to some extent in the flatter tails of the synthetic orange histogram with less dense probability density around zero. The positive trend of $\bar{C}\left(t\right)$ is not modelled in the simulated time series *x*.

**Table 1 entropy-25-01257-t001:** Comparing the predictions of the naive forecast ${\hat{y}}_{t+1}=y_{t}$, the Langevin equation (LE) and the GLE with memory kernel length k=3 via their $\rho^{2}$ score on in-sample training data and out-of-sample test data (α% training data and the last 1−α% of the time series as test data). Note that the partially negative $\rho^{2}$ for the test data indicates that the test data is quite difficult to predict, which corresponds to the high fluctuations in the final part of the time series in Figure 1.

Model α	In-Sample			Out-of-Sample		
	80%	85%	90%	80%	85%	90%
Naive	0.07	0.15	0.19	−0.21	−0.15	−0.11
LE	0.29	0.32	0.35	−0.14	−0.01	0.08
GLE	0.39	0.42	0.45	0.01	0.08	0.10

## Data Availability

All data are publicly available and the raw data can be accessed via the package [33]. The preprocessed time series is available via [38].

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
