# Peer review of "Memory Effects, Multiple Time Scales and Local Stability in Langevin Models of the S&P500 Market Correlation"

_entropy, 2023, doi:10.3390/e25091257_

Round 1
Reviewer 1 Report
The authors have carried out a substantial piece of work. I like both the idea and the presentation. The main audience will be physicists and finance researchers interested in the particular method applied here; hence a good fit for Entropy.
My main criticisms have to do with the chosen time period and the selection of stocks:
1. Why do you select 1992-2012 as the time period studied?
2. "Only stock data of companies that were part of the S&P 500 index during at least 99.5% of the time were used for this analysis." Why does this not introduce a selection bias?
3. Can one use data from the period 2013-2022 to rerun the analysis and check for, e.g., robustness of results?
There are a few places where wording can be improved. Here an example
"Because the economy is a system with a large number of interacting parts (traders and companies)"
Do not start a sentence with "because", and the economy has even more actors than those listed.
I therefore stated that minor editing is required.
Author Response
Dear reviewer,
we would like to thank you for your quick and valuable feedback. Please find our response in the attached pdf where we describe the changes to our manuscript.
Kind regards,
Tobias Wand on behalf of all authors

Reviewer 2 Report
The authors delve into the intricate dynamics of market correlations, particularly in the context of the S&P500. The authors emphasize the significance of understanding memory effects in market correlations. By fitting the data to a GLE, they uncover a notable memory effect that spans at least three trading weeks. This memory effect enhances forecasting accuracy, underscoring its importance in optimal portfolio selection and risk management. The use of Bayesian resilience estimation further enriches the analysis, suggesting the existence of hidden slow time scales in market data. Overall, the study provides valuable insights into the nuanced behaviors of market correlations, advocating for a more comprehensive approach to understanding and predicting market dynamics.
The paper is well-written and can be published as is.
Author Response
Dear reviewer,
We would like to thank you for your fast and positive feedback to our manuscript.
Kind regards,
Tobias Wand on behalf of all authors

Reviewer 3 Report
Possibly the paper is very useful for practitioners, but mathematical level is very low.
Good.
Author Response
Dear reviewer,
We would like to thank you for your review. Please find attached our response.
Yours sincerely,
Tobias Wand on behalf of all authors

Round 2
Reviewer 1 Report
I am happy with the revised manuscript.
Reviewer 3 Report
Mathematical component of the paper is the same fantastic as before. First, this is NOT a Langevin equation, neither (5) nor (6). Ok, the name is not very important. But the authors operate with white noise not operating with it. "Independent Gaussian noise"--it is out of mathematical sense, independent of what? "First, we estimate a Markovian Langevin equation"-it is out of sense, it is impossible to estimate equation. Expansion (7)-(8) is fantastic, because equation with such coefficient can have no solution. In general, the question of existence and uniqueness of solution does not considered at all, so, the authors study and "estimate" something that can be non-existing. The "feedback"of the authors is absolutely formal.